# Analysis of Flow and Land Use on the Hydraulic Structure of Southeast Mexico City: Implications on Flood and Runoff

Rosanna Bonasia [1,*,†], Lorenzo Borselli [2,†], Paolo Madonia [3,†]

1    Tecnologico de Monterrey, School of Engineering and Sciences, Carretera Lago de Guadalupe Km 3.5 Atizapán de Zaragoza Col. Margarita Maza de Juarez, Cd López Mateos 52926, Mexico

2    Instituto de Geologia, Facultad de Ingenieria, Universidad Autonoma de San Luis Potosí (UASLP), Av. Dr. Manuel Nava 5, San Luis Potosí 78290, Mexico; lorenzo.borselli@uaslp.mx

3    Istituto Nazionale di Geofisica e Vulcanologia, Sezione di Catania—Osservatorio Etneo, Piazza Roma 2, 95125 Catania, Italy; paolo.madonia@ingv.it

*    Correspondence: rosanna.bonasia@tec.mx

†    These authors contributed equally to this work.

**Abstract:** The southeast of Mexico City is one of the last areas of environmental importance for the region. However, rapid urban expansion has led to a runoff increase in the presence of intense rainfall. This situation is common to many peri-urban centers close to large cities, where the urbanization of previously green areas has had a direct negative influence on the hydraulic structure. This work proposes a study that combines hydrological analysis for the definition of precipitation scenarios with hydrodynamic simulations based on the current land use. Reconstructed flood scenarios show that the runoffs descending from mountainous areas flow into cemented channels with hydraulic sections and characteristics not adequate to drain specific discharges that can reach $0.90 \text{ m}^2/\text{s}$ and water depths of the order of 2 m, caused by extreme weather phenomena, determining flooding in nearby areas. Runoffs are also intensified by the presence of non-urbanized open spaces in a state of abandonment, whose soil does not favor infiltration and promotes the flooding of residential centers with water levels higher than 1 m. The results indicate an urgent need to adopt actions to reduce flooding and favor infiltration in an area of the city that is also important for aquifer recharge.

**Keywords:** flood modeling; conservation land; Iber software; numerical models; Mexico City; land use



## 1. Introduction

The southern area of Mexico City is affected by recurring flood phenomena, which are in part due to its topographic location, downstream of a mountain system, as well as in part, and probably mainly due to incorrect urban planning that has led to drastic and unfavorable changes in land use. In a study by Zuñiga and Magaña [1], in which a methodology for estimating flood risk in Mexico associated with heavy rains by considering changes in land cover and land use is developed, the authors find that deforested and urbanized areas are the physical and social factors that lead to the deterioration of hydrographic basins and to greater vulnerability to heavy rains, which occurs particularly in the center and south of the country. Over the past 50 years, large metropolises in Mexico have been the scene of uncontrolled urban sprawl, and ineffective urban management has led to the marginalization of areas that are most affected by flood-related problems [2].

The intense urban expansion that characterizes Mexico City has led to substantial changes in land use, even in districts that represent ecological conservation zones, leading to the formation of areas defined as "peri-urban" or "rur-urban", distributed in the vicinity of the metropolitan core [3]. One of the southern delegations of the city that has suffered the most from the effects of urban expansion is Milpa Alta. This area is occupied, for about the entirety of its territory, by conservation land, but despite this, the expansive,

dispersed urban growth has caused drastic changes in land use (forestry-agricultural, forestry-urban, agricultural-urban), especially in the mountainous zone and the agricultural areas surrounding rural towns [4]. According to a study carried out by the Environmental and Land Management Office of Mexico City [5], in the period of 2000–2010, changes in vegetation cover within the region, related to micro-watersheds and landscape units, generated changes in the infiltration and runoff processes of the region. In particular, the urbanization processes have modified the hydrological response of the catchment areas, due to urbanization in areas with high slopes, which leads to an alteration of the natural drainage networks and an increase in impermeable areas on the surface. To prevent land use change and deal with environmental crimes affecting Milpa Alta, in 2020, the Ministry of the Environment (SEDEMA) of Mexico City, through the General Directorate of the Commission for Natural Resources and Rural Development (CORENADR), filed a complaint with the Federal Attorney for Environmental Protection (PROFEPA) [6]. The degradation of the forest cover entailed the loss of aquifer recharge zones, and the change in land use led to the incidence of dangerous phenomena that, in agreement with the Delegational Urban Development Program of the Federal District [4], are grouped into the following main categories: subsidence, landslides, soil erosion, ground cracking, and, of course, floods.

Since the 1930s, Milpa Alta has been affected by intense rain flood phenomena that have caused extensive damage to properties, crops, and people. In 1935, the town of San Pedro Atocpan, suffered one of the greatest tragedies in all of Milpa Alta. On 3 June 1935, an intense precipitation of an extraordinary nature strongly lashed out at the residents of San Pedro Atocpan while the town festival was being celebrated. Merchants and visitors were swept away by rough waters that came down from the mountain, and the following day, 150 deaths were counted [7]. In 1998, another storm caused the overflow of the ravine located in the town of San Lorenzo Tlacoyucan, causing three deaths [8]. In 2004, an intense rain caused damage similar to that of the 1990s, except for the loss of human lives, fortunately. During this event, the water flowed through one of the ravines that passes through the delegation, the Barranca Seca, and overflowed, generating floods in San Antonio Tecómitl, reaching 1 m in depth. The material losses only affected homes irregularly settled on the conservation land [9].

Mexico City is experiencing a paradox related to water resources. While it is necessary to dislodge large volumes of water from rain in a short time, there are, on the other hand, deficiencies in the supply of drinking water, and this depends, among other causes, on the fact that aquifers are over exploited and their recharge is complicated by the presence of soils that do not favor water infiltration. This is precisely the case of Milpa Alta and adjacent territories. In recent years, members of the agrarian nucleus of Milpa Alta have demanded better water service and the cancellation of their payment, based on the argument that the water is recharged in the forests that are part of the communal territory. However, the scarcity of water suffered by the population is due to depletion of the groundwater table, making necessary the use of tank trucks for integrating available water resources [10].

In this work, we analyze the current hydraulic structure of the study area, on the basis of its land use characterization, and we study the effects it has on runoff in inhabited centers and rural areas. The working hypothesis is that the site under study, despite being characterized by soils with a medium to high water infiltration capacity, due to an unfavorable land use change, has considerably lost this capacity, favoring an increase in surface runoff. This can have a potential negative effect on flood levels and on the aquifer recharge as well.

The problem of the relationship between runoff and the recharge of the aquifer is complex and has been treated under different aspects. For example, in semi-arid regions where water resources are scarce, the possibility of increasing water resources by recharging the aquifer with surface runoff infiltration has been investigated [11], with a limitation, however, due to the presence of sediment runoff that involves the need for very efficient maintenance of the managed recharge structures of the aquifer. Moreover, despite the

role of land use change and the potential effects on the generation capacity of aquifer runoff and recharge having been addressed by several authors, the role of different vegetation covers does not provide a clear and unambiguous effect. Bellot et al. [12] indicate that deforestation can lead to a reduction in the recharge capacity of aquifers; Zuazo and Pleguezuelo [13] indicate that vegetation cover is essential for protection from soil erosion; and Descheemaeker et al. [14] present the role of natural vegetation in semi-arid environments in the production of runoff. The complex dynamics of land use and different vegetation types has also been addressed by Ghiglieri et al. [15], Steinel et al. [11], and Eshtawi et al. [16]. Based on the evidence from this body of literature, which indicates the extreme complexity of the relationship between runoff management and the criticality in aquifer recharge, the considerations of the influence that a change in land use in the study area can have on the recharge of underground water will be of a qualitative nature. On the other hand, calculated runoff values in terms of water depth and specific discharge rate are provided with the aim of quantitatively analyzing the hydraulic structure of the area in its current condition and by proposing changes in land use. The calculation of runoff and susceptibility to flooding has undergone numerous improvements in recent years with techniques that have reached acceptable levels of accuracy. For example, a method has recently been developed to predict future waterlogging areas in urban zones, coupling maximum entropy with the Future Land Use Simulation model in order to accurately predict a spatio-temporal model of land use [17]. Another recent study aims to model flooding in river basins by applying maximum entropy and frequency ratio methods, as well as analyzing the relationship between flood events and factors affecting flood risk [18]. Data-driven techniques are also gaining widespread use for long- and short-range forecasting, with the application of machine learning techniques [19]. Additionally, the application of the numerical analysis of flood scenarios in urban areas has reached, in the last decades, high levels of precision for calculating accurate runoff and peak discharge rates as a function of urbanization forecasts [20–22].

The problem addressed in this work concerns the situation of many peri-urban areas close to large cities, not only in developing countries, such as Mexico. The increase in urbanization of these zones, previously occupied by green areas, has caused a direct influence on the increase in surface runoff [23,24]. Therefore, the methodology presented in this work can provide the basis for a correct analysis of the hydraulic structure of the study area and can provide the basis for a correct territorial planning. The applied methodology consists of three basic steps: (i) the hydrological study for the construction of precipitation scenarios corresponding to different return periods; (ii) the application of a hydraulic model for the calculation of surface runoff levels with the Iber software [25], based in the analysis of current land use; and (iii) the soil use analysis and improvement of infiltration parameters in the hydraulic simulations.

## 2. Materials and Methods

### 2.1. Description of the Study Area

The study area chosen for this work mainly includes the Milpa Alta delegation, its northeast border with the Tlahuac delegation, and the border to the east with the State of Mexico (Figure 1). The particular attention on Milpa Alta lies in the fact that this area is an important nature reserve of the city, due to its physiographic conformation and the presence, despite extensive urbanization, of a dense vegetation cover. Furthermore, the area under consideration has been affected over the years by floods, which have brought considerable disruption to the communication infrastructures and to daily life.

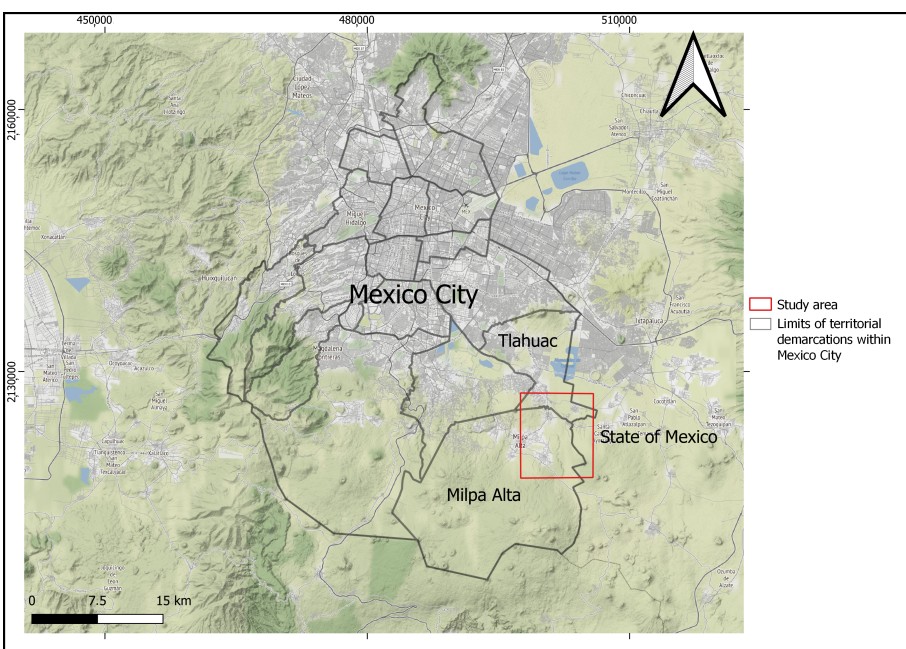

**Figure 1.** Location map of the study area, delimited by the red line, southeast of Mexico City, within the Milpa Alta delegation, Tlahuac, and the State of Mexico.

Milpa Alta is 1 of the 16 districts of Mexico City, located in the foothills of the Ajusco-Chichinauhtzin mountain range, which separates the state of Morelos and the Mexican capital. It has formally been a delegation since 1929 and is part of the so-called Metropolitan Zone of Mexico City. With an area of 228 km$^2$, it is the second largest delegation and constitutes an important nature reserve in the center of the country. The delegation is located in one of the forest areas far from the urban center, characterized by the presence of fir and pine forests at the foot of the mountainous area to the south. Due to its location in a natural area, 100% of its territory is listed as conservation land, that is, an area that, due to its ecological characteristics, provides environmental services necessary for the maintenance of the quality of life for Mexico City's inhabitants. Although the denomination of conservation land imposes limits on the delegation's urbanization, currently, 10% of the territory is occupied by twelve rural towns. The rest of the delegation surface has 41% dedicated to the development of agricultural activities, and the remaining 49% is occupied by wooded areas, which represent environmentally protected zones [26]. Due to the peculiar physiographic and geological characteristics of the territory under study, this is an area of fundamental importance for the recharge of the Mexico City aquifer, but, at the same time, of intense water extraction.

According to the statistical notebook of the National Institute of Statistics and Geography (INEGI) [27], more than 96% of the Milpa Alta surface, specifically, is the product of Quaternary geological activity. The study area occupies the southern sector of the Basin of Mexico, characterized by volcanic deposits from the Ajusco volcano and the Chichinautzin mountain range, which were formed at the end of the Tertiary period and mainly during the Quaternary [28,29]. A large part of the delegation territory is mainly constituted of rocks of volcanic origin, which can be divided into five geological units: basic extrusive igneous (Quaternary age), pyroclastic deposits (Neogene period), basic extrusive igneous (Neogene period), acidic extrusive igneous (Quaternary age), and alluvial and recent sedimentary deposits of the Quaternary (Figure 2).

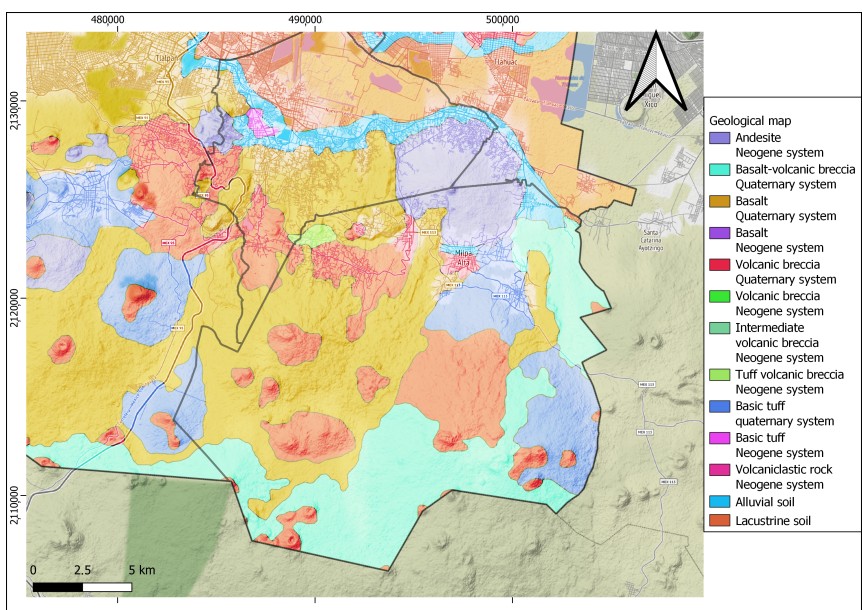

**Figure 2.** Geology map of the Milpa Alta delegation and surrounding area. After [30].

In the center, south, and west of the study area, the dominant soil type is andosol (63%), and in the north and east there are feozem class soils (17%), which are used for planting crops of corn, beans, nopal, or fruit trees. These soils are highly sensitive to erosion [31]; they are basically made up of lava and scoria fragments, agglomerate, and pyroclastic deposits of various sizes, surrounded by a matrix of sand, silt, and clay-sized sediments. These deposits, due to their high permeability, make the area one of the main aquifer recharge zones in the basin.

The study area is located within the aquifer of the Metropolitan Area of Mexico City, which falls into the valley of Mexico basin. This is an endorheic basin of a lacustrine nature, where the crust has undergone great stress, producing intense fracturing. It is surrounded by mountains and covered at different points by lacustrine areas resulting from lakes that existed at the end of the ice age. When the basin closed in the Upper Quaternary, the pluvial waters were confined, forming a set of shallow lakes. The aquifer supplies water to Mexico City and the urban area that basically depends on the availability of this aquifer for the supply of drinking water. In general, the city and the aquifer are separated, for the most part, by a clayey aquitard, with a thickness of around 50 m. The aquifer lays at depths greater than 800 m, and the extraction wells have depths between 100 and 400 m. In the urban center of San Antonio Tecómitl, north of the Milpa Alta Delegation, more than 10 water extraction wells have been drilled and purification plants have been installed.

For the most of the delegation, that is, on the slopes of the mountains, the climate is semi-cold sub-humid, with abundant rains in the summer. This area is practically uninhabited and covered by pine and fir forests. The remaining part has a sub-humid temperate climate with summer rains and is the territory where most of the population settles and where the most important economic activity of the delegation is practiced [26].

### 2.2. Hydrological Study for the Definition of Precipitation Scenarios

The precipitation intensity scenarios for the study area were calculated on the basis of the construction of intensity–duration–frequency (IDF) curves, where the probability of occurrence of intense precipitation was characterized by defining four return periods (10, 50, 100, and 500 years). For the construction of the IDF curves, the empirical approximation of Aparicio [32] was applied in this work:

$$I = \frac{kT^m}{(t+t_0)^n},$$ (1)

where $I$ is the maximum precipitation intensity (mm/h); $t$ is the rain duration (min); $T$ is the return period (years); and $k$, $m$, $n$, and $t_0$ are constants that are calculated by linear regression analysis.

The precipitation values used to calculate rain intensities were obtained from the hydrometherological stations closest to the study area, whose database is available at the National Surface Water Data Bank (BANDAS) [33]. The stations with influence on the study area are the following: 9051 Tlahuac, 9032 Milpa Alta, and 9045 Santa Ana Tlacotenco, for which annual precipitation values are available from the 1960s to 2010. The average annual precipitation was calculated for each station as

$$\overline{h_p} = \frac{\sum_{i=1}^{n} A_i h_{pi}}{A_T}, \tag{2}$$

where $h_p$ is the precipitation height corresponding to the $i$ station, $A_i$ is the area of influence of the $i$ station, and $A_T$ is the total area of the basin, which, in the case of the study area, is equal to 4265 km$^2$. The area of influence of the hydrometeorological stations was calculated using the Thiessen polygon method [34]. The characteristics of each station, together with the calculated average rainfall heights, are shown in Table 1.

**Table 1.** Characteristic parameters of each hydrometherological station and calculated values of average annual precipitation heights.

| Station | 9051 (Lon: −99.004 Lat: 19.263) | 9032 (Lon: −99.0219 Lat: 19.1906) | 9045 (Lon: −99.0028 Lat: 19.1789) |
|---|---|---|---|
| $h_{pi}$ (mm) | 11.008 | 9.501 | 8.383 |
| $A_i$ (km$^2$) | 0.628 | 1.507 | 2.129 |
| $\overline{h_p}$ (mm) | 1.621 | 3.358 | 4.185 |

The maximum daily precipitation rates for each return period and different rain durations were calculated with the Gumbel method [35], which is a logarithmic normal distribution according to which the extreme value is obtained from the frequency analysis. Rainfall intensities were obtained as the ratio of rainfall depths falling during a given period to the storm duration. To obtain the rainfall intensity values dependent on each return period and on the duration of the event, Equation (1) was linearized through the following logarithmic transformation:

$$\log I = \log k + m \log T - n \log t + t_0, \tag{3}$$

A linear regression was applied to the rainfall values of the three stations in order to obtain the following values of the equation constants: $k$ = 14.668, $m$ = 0.084, $n$ = 0.616, and $t_0$ = 0. Figure 3 shows the IDF curves for the selected return periods and a storm duration of 120 min.

The calculated rainfall intensity values were used as inputs for the flood scenario simulations that are described in the following paragraph.

### 2.3. Flood Scenarios Modeling

With the aim of calculating water depths affecting the study area as a consequence of surface runoffs due to different precipitation scenarios, flood simulations were carried out with the Iber software [25]. The software is a two-dimensional model for the simulation of free surface flow, morphodynamics, transport processes, and habitat in rivers and estuaries, which solves the shallow water equations in 2 dimensions using an explicit unstructured finite volume solver [36]. The model has been extensively validated and applied to flood studies [37–39] as well as to modeling of rainfall runoff processes [40,41]. In the Mexican territory, the Iber software was applied for the analysis of the influence of land use change

on the increase in flood levels [42] and for the identification of potential overflow points of a river with consequent flood scenarios in the alluvial plain [43].

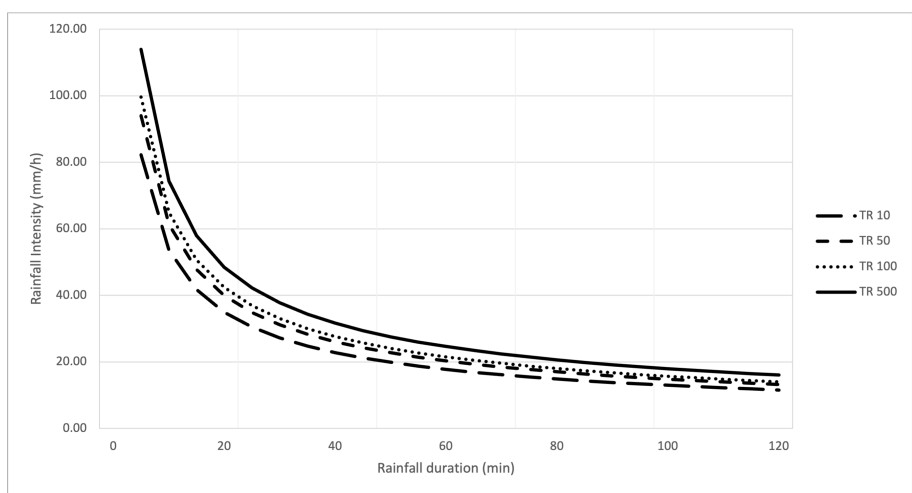

**Figure 3.** Intensity–frequency–duration curves for the return periods TR 10, 50, 100, and 500 years.

The computation domain considered for the hydraulic simulations occupies an area of approximately 79,400 km² and is shown in Figure 4, delimited by the red line. The figure also shows the distribution of land uses as they are classified by INEGI and available in its database [44].

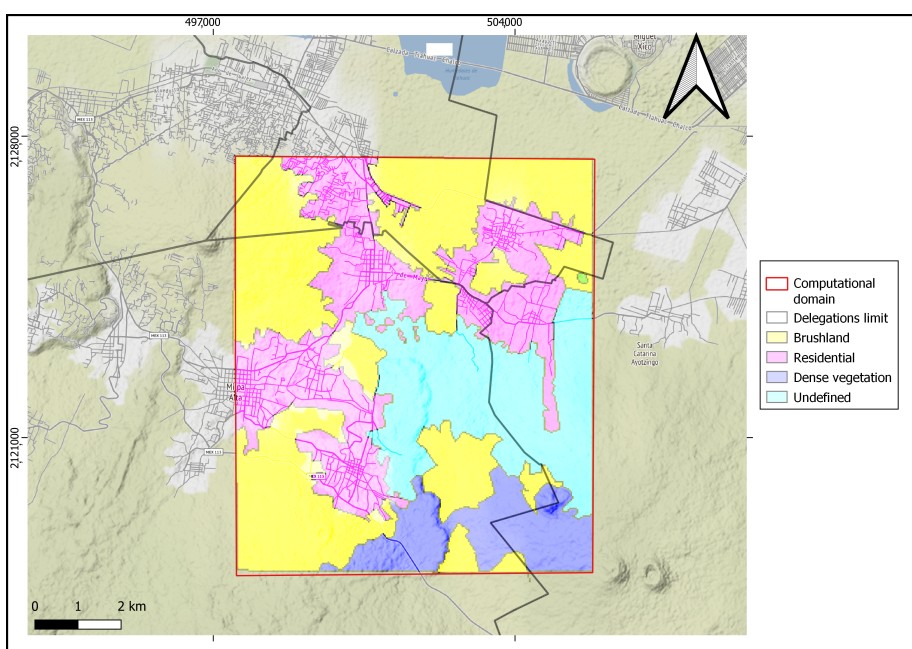

**Figure 4.** Location map of the computational domain and land-use distribution.

According to data reported by INEGI, there are four recognizable land uses in the area under study: brushland, residential, dense vegetation, and "undefined", to which the following Manning coefficients have been assigned for the characterization of surface roughness: 0.05, 0.15, 0.18, and 0.032, respectively, which are established on the basis of the River Engineering Manuals [45]. The study area was discretized with a triangular unstructured mesh to obtain different levels of spatial resolution in the different areas of the domain, generating a mesh of 13,261 elements, with element sizes of 10 m in urban areas and 30 m in the rest of the domain. Three-dimensional geometry has been constructed using a terrain-type digital elevation model (DEM) with 5 m resolution derived from satellite and

airborne remote sensing. The DEM, available in the INEGI database [46], describes a recent conformation of the territory (2018 model edition), and its resolution allowed the geometry to be constructed in 3 dimensions with enough detail to appreciate the presence of natural channels and ravines, as well as urban buildings and structures.

In order to take infiltration losses into account, the Iber software allows for the application of an infiltration model. In this work, the Curve Number model of the Soil Conservation Service [47] was applied. This model is a well-known formulation used in a large variety of hydrological studies, according to which an initial loss is followed by a variable infiltration capacity that decreases approximately exponentially over time. The application of the model depends on the use of only one value, the curve number, which depends on the slope of the land, the type, and the land use. Based on the soil and geological characteristics previously described, it can be considered that the study area has a moderate to high infiltration capacity, being made up of moderately well-drained to well-drained soils, due to the presence of fine to moderately coarse textures. These characteristics make the study area fall into the hydrologic soil group B [48]. Then, residential soil, brushland, and dense vegetation, observable in Figure 4, have been assigned curve numbers 80, 67, and 32, respectively. A separate discussion must be made for the soil that INEGI classifies as "undefined". A field patrol made it possible to observe that the area is mainly covered by spontaneous vegetation (Figure 5a), barren areas (Figure 5b) affected by the combustion of shrubs and waste of various kinds, interspersed, mainly to the north, with nopal cultivations (Figure 5c). These characteristics give the soil a poor infiltration capacity, for which it was assigned a curve number of 80.

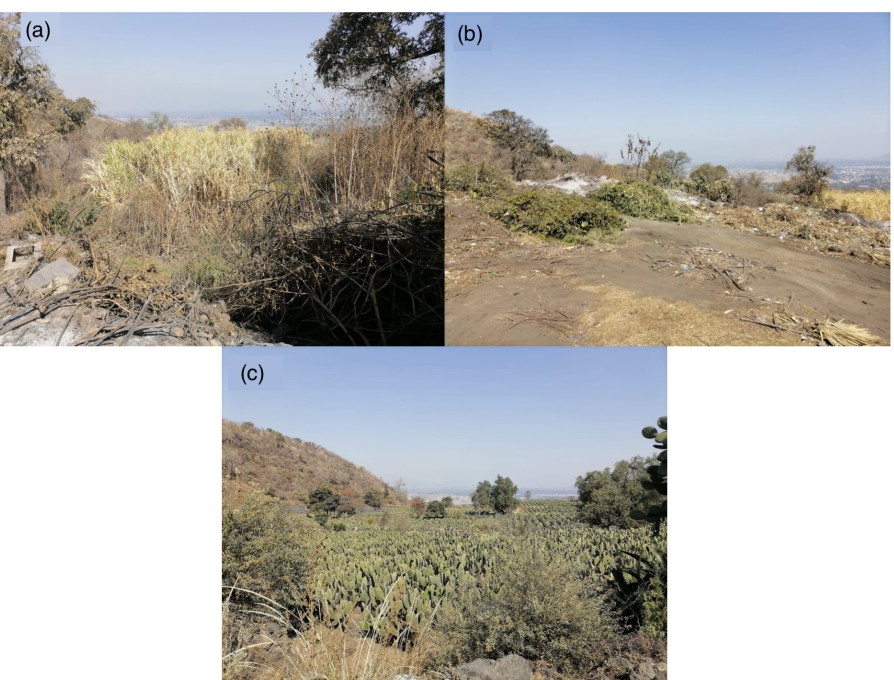

**Figure 5.** Photos of the "undefined" soil classified by INEGI with the following characteristics: (**a**) spontaneous vegetation; (**b**) barren areas; (**c**) nopal cultivation.

Table 2 summarizes the Manning coefficients and curve numbers assigned to the soils identified in Figure 4.

Initial conditions of water depth equal to zero were assigned because there are no permanent water channels in the area. Table 3 shows the numerical parameters for the flood scenario simulations.

**Table 2.** Manning coefficients and curve numbers assigned to the land uses recognized in the study area.

| Land Use | Manning Coefficient | Curve Number |
|---|---|---|
| Brushland | 0.05 | 67 |
| Residential | 0.15 | 80 |
| Dense vegetation | 0.18 | 32 |
| Undefined (bare land) | 0.032 | 80 |

**Table 3.** Numerical parameters for hydraulic simulations.

| | |
|---|---|
| Maximum simulation time (s) | 7200 |
| Results interval (s) | 60 |
| Numerical scheme | Hydraulic—first order |
| Courant–Friedrichs–Lewy number | 0.45 |
| Dry/Wet limit | 0.01 |
| Total calculation time | Min 15 min (Tr 5 simulation)–Max 23 min (Tr 500 simulation) |

The maximum simulation time corresponds to the rainfall duration in the IDF curves (120 min). The Courant–Friedrichs–Lewy number (0.45) is a dimensionless number used to analyze numerical stability and represents the ratio between the size of the time step and the size of the spatial step of the numerical method used. Finally, the dry/wet limit (0.01) defines the tolerance assumed to define a cell of the computational domain as dry and therefore not considered in the calculation. The input hydrological parameters of the model are represented by the values of rainfall intensity calculated for the different return periods, as described in Section 2.2. Simulation results are calculated values of water depth and specific discharge rates for the different return periods.

## 3. Results

*Flooding Maps*

Figures 6–9 show the results of the numerical simulations of flood scenarios in terms of water depths calculated for the return periods of 10, 50, 100, and 500 years, respectively.

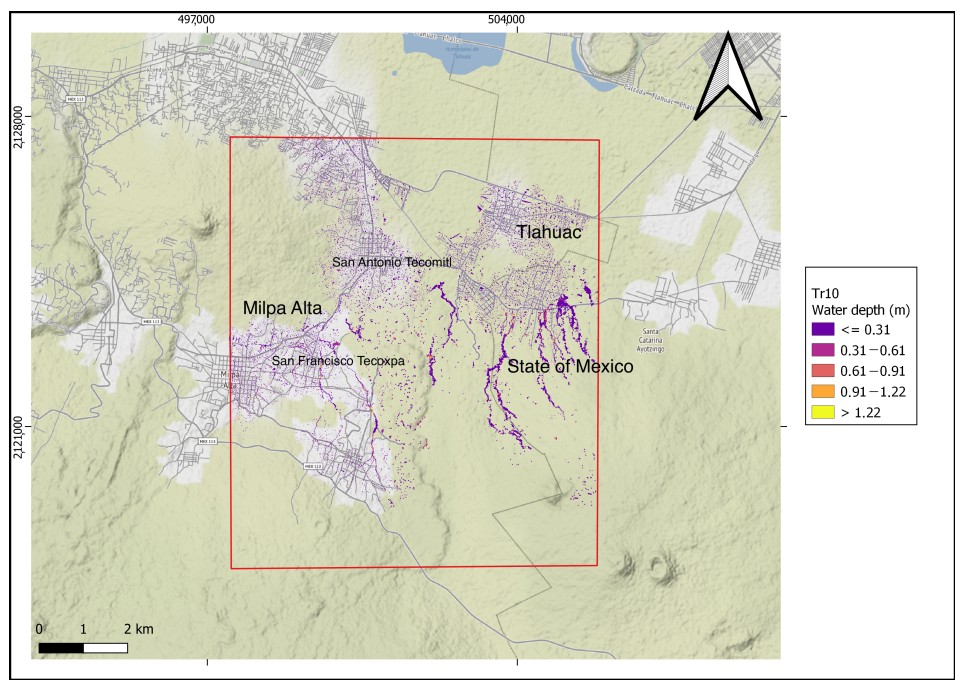

**Figure 6.** Result of simulations of inundation water depths for the return period 10 years.

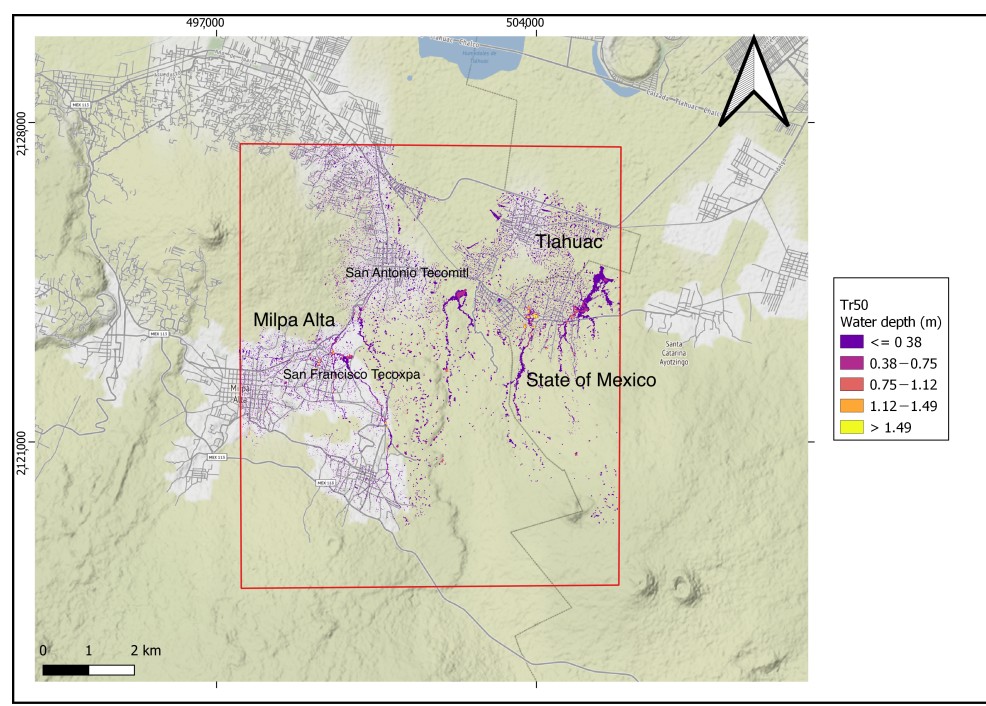

**Figure 7.** Result of simulations of inundation water depths for the return period 50 years.

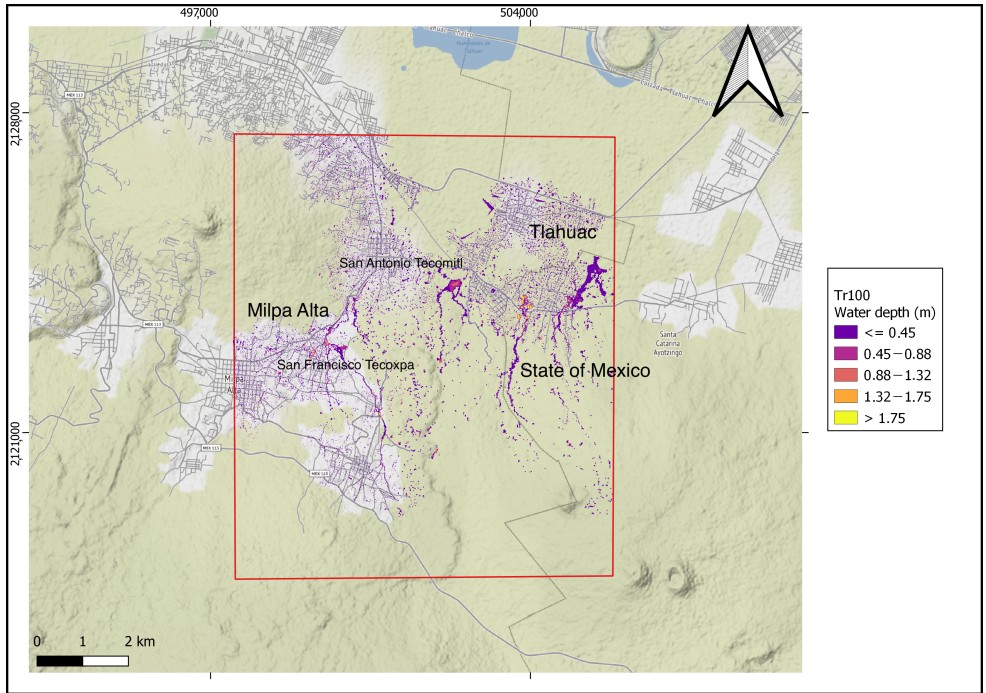

**Figure 8.** Result of simulations of inundation water depths for the return period 100 years.

The maps show that with relatively frequent rains (10- and 50-year return periods in Figures 6 and 7), floods occur with water depths averaging between 30 and 70 cm. The inundated area is more extended in the 50-year return period scenario, and the deepest water depths (>1.49 m) affect the inhabited center on the border between the Milpa Alta delegation, Tlahuac, and the State of Mexico. High water depths are also observed in San Francisco Texcopa, an urban center particularly affected by the 1998 flood.

The situation worsens when simulating fewer frequent rain events, corresponding to return periods of 100 and 500 years. Results of these simulations show that the maximum flood levels exceed 1.75 m for the 100-year return period (Figure 8) and 2.16 m for

the 500-year return period (Figure 9) in the populated centers mentioned above. These water depths are compatible with the effects of floods caused by extreme meteorological phenomena that have occurred over the years, which caused floods greater than 1.50 m, particularly in the inhabited centers of San Antonio Tecómitl, Villa Milpa Alta, and San Francisco Tecoxpa, as reported by local news.

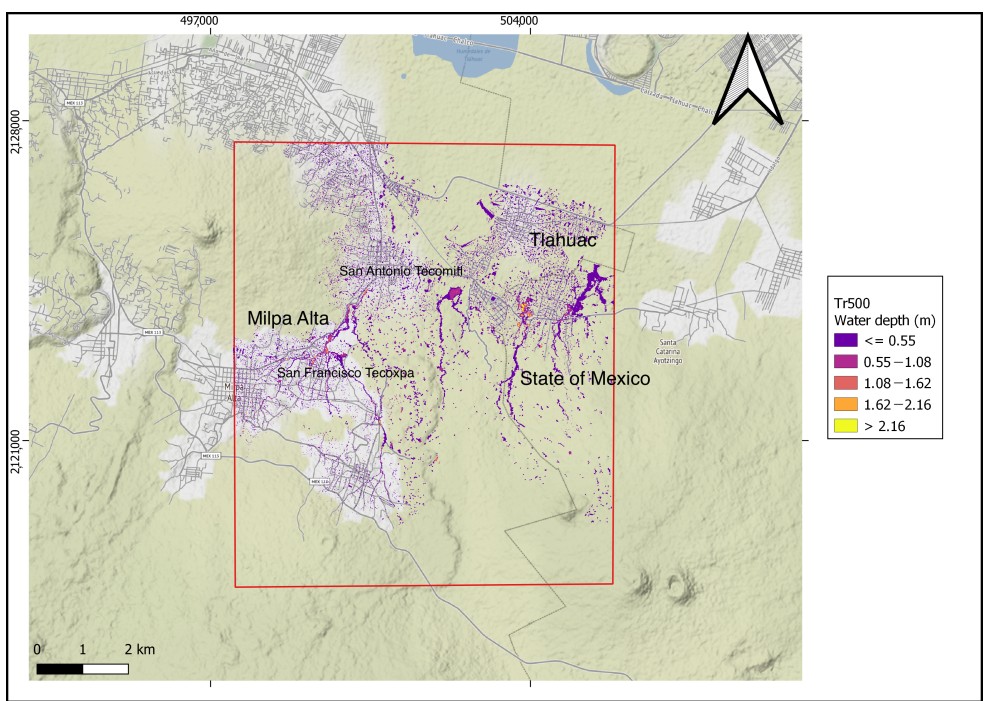

**Figure 9.** Result of simulations of inundation water depths for the return period 500 years.

From maps related to all return periods, it can be seen that the main water collector channels towards the inhabited centers are natural canyons that descend from the mountains to the south of the study area. In the municipality of San Antonio Tecómitl, one of these channels takes the name Barranca Seca and runs through the town. The channel is cemented, and its depth varies between 50 cm and 1.50 m. Simulation results also provided specific discharges, which describe the variation over time of both water velocity and depth as a linear function (discharge $q = hu$ [m$^3$/s/m]) [49,50]. Figure 10 shows a distribution map of the specific discharges calculated for the 500-year return period. The detail on a portion of the Barranca Seca (Figure 11) shows that discharge reaches values between 0.63 and 0.94 m$^2$/s.

In the central and eastern portion of the computing domain, the flow crosses the area occupied by the land use characterized as "undefined" by INEGI (shown in Figure 4), with water depths ranging from 30 cm to more than 50 cm, depending on the considered return period. As previously described, this area is mostly in a state of neglect, covered by soils that do not favor water infiltration. However, the northern portion of this area, bordering the residential land, is occupied by nopal farms. According to Vázquez-Méndez et al. [51], the fertility of nopal islands produces very low maximum runoff values and is effective in attenuating the erosive forces of rain and runoff. To investigate the effect that a land use change might have on runoff, we considered the possibility that this undefined patch of land can be almost completely covered by crops such as nopal, which favor infiltration by decreasing runoff. For this purpose, an additional simulation was performed, for the highest return period, by changing the "undefined" soil curve number from 80 to 67. Figure 12 shows the result of this simulation, highlighting the effect of the hydraulic structure changes: the runoff in the central part of the "undefined" soil disappears, and the flooding in the downstream residential area decreases drastically.

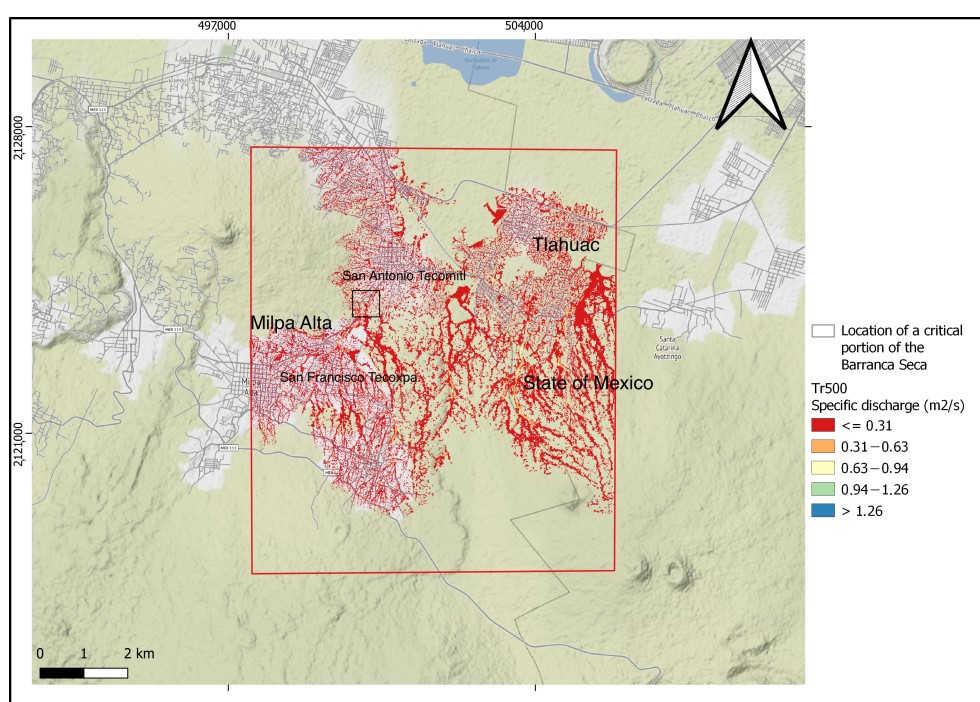

**Figure 10.** Simulation results of specific discharge for the return period of 500 years. The black box indicates the location of the portion of Barranca Seca where high discharge values are detected, which are shown in Figure 11.

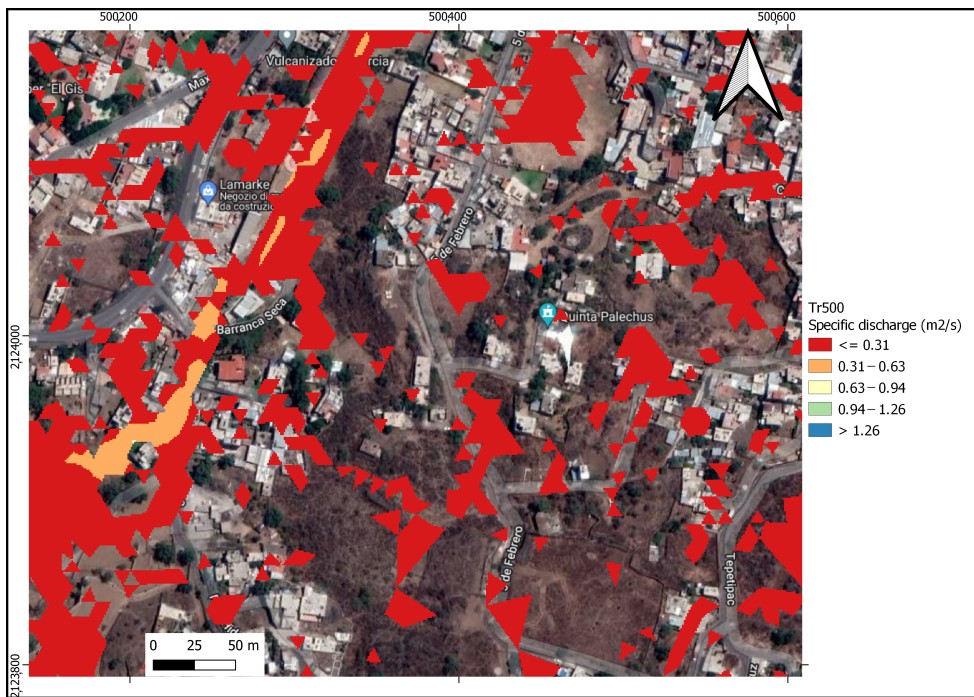

**Figure 11.** Specific discharge values calculated for the return period of 500 years in San Antonio Tecómitl at the Barranca Seca.

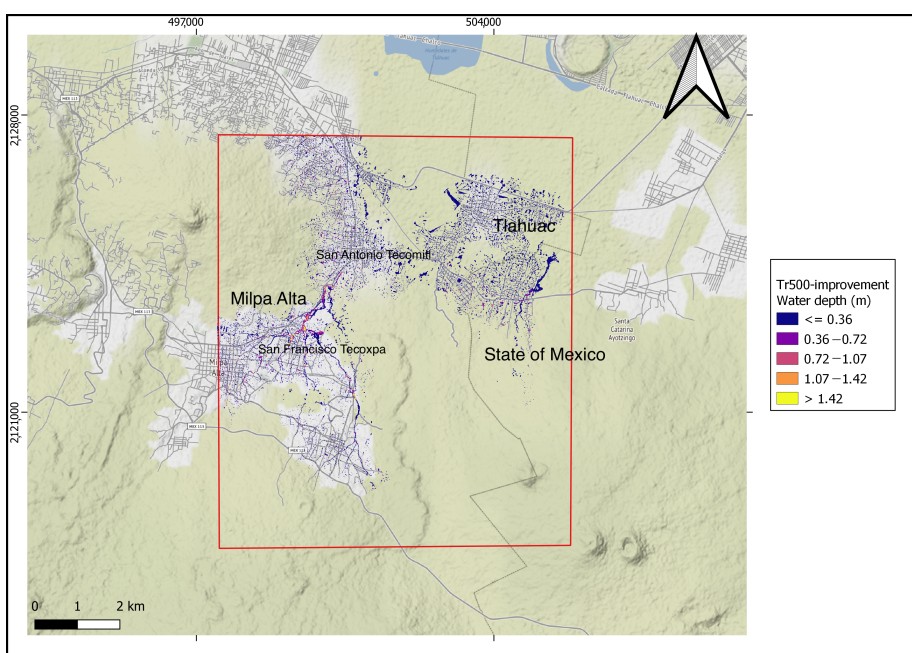

**Figure 12.** Simulation results of water depths for the return period of 500 years, obtained by improving the curve number of the "undefined" soil type.

## 4. Discussion

The Milpa Alta delegation and the adjacent territories occupy one of the few conservation lands in Mexico City, considered such because they are characterized by dense vegetation, essential for the ecosystem of the city, and important for the recharge of the aquifer that feeds a large part of the Valle de México area. However, despite the limitations imposed by the territorial planning legislation, urbanization has had an uncontrolled increase, due to the expansion of irregular settlements, which has caused a drastic change in land use not only in terms of expansion of urban areas, but also in terms of deforestation and abandonment of rural areas not yet affected by urbanization. The occupation of conservation lands in Mexico City is a process that has taken place since the 1970s [52] and also affects the Milpa Alta delegation with a significant increase since 1985 when, following the earthquake that affected the city, the population began to migrate to places further away from the center that had suffered the greatest damage [53]. These events have determined that, in terms of the geographic distribution of the runoff rate by hydro-ecological units, between the years of 2000 and 2010, there have been significant changes, particularly in Milpa Alta, according to the runoff analysis carried out by the Environmental Attorney of Mexico City [5].

The results of the hydrodynamic simulations carried out in this work, induced by precipitation scenarios corresponding to events with different probabilities of occurrence (return periods from 10 to 500 years), provide important information on the hydraulic structure of the study area. In the case of more frequent events (Figures 6 and 7), the current condition of land use determines runoffs that can generate flood levels ranging from about 40 cm up to maxima >1.49 m. The runoffs particularly affect the eastern portion of the computing domain, on the border between the Milpa Alta delegation and the State of Mexico and between Milpa Alta and Tlahuac. Runoffs smaller than 40 cm are ephemeral; however, water depths exceeding one meter can be observed in the residential portions north of the study area. These stagnations occur mainly in the municipality of San Juan and San Pedro Tezompa (State of Mexico), which is listed among the most susceptible to flooding in this area, according to the General Coordination of Civil Protection of the State of Mexico [54].

However, one factor that stands out from the results of the hydrodynamic study concerns the effect that particularly heavy rainfall can have on the cemented channels

present in the inhabited centers in the western portion of the study area. As shown in the results, in the San Antonio Tecómitl urban site, one of these canals (the Barranca Seca shown in Figure 13) crosses almost the entire inhabited center, and, in certain portions (the area bounded by the black line in Figure 10), it does not have a hydraulic section adequate to drain the flow of water that descends from the natural canyons upstream of the area under study. In fact, in this critical area, the channel has a width of about 2 m and a depth of 1.50 m. If we consider the results of the simulations for the 500-year return period, we can see that the maxima observed in the area highlighted in Figure 10 are equal to 2.16 m for the water depth and 0.94 $m^2$/s for the specific discharge, which leads to flow rate values in this area of approximately 2 $m^3$/s. This value is at the limit of the channel's critical flow rate, above all considering that its section is reduced by the presence of waste and spontaneous vegetation, as can be seen in Figure 13.

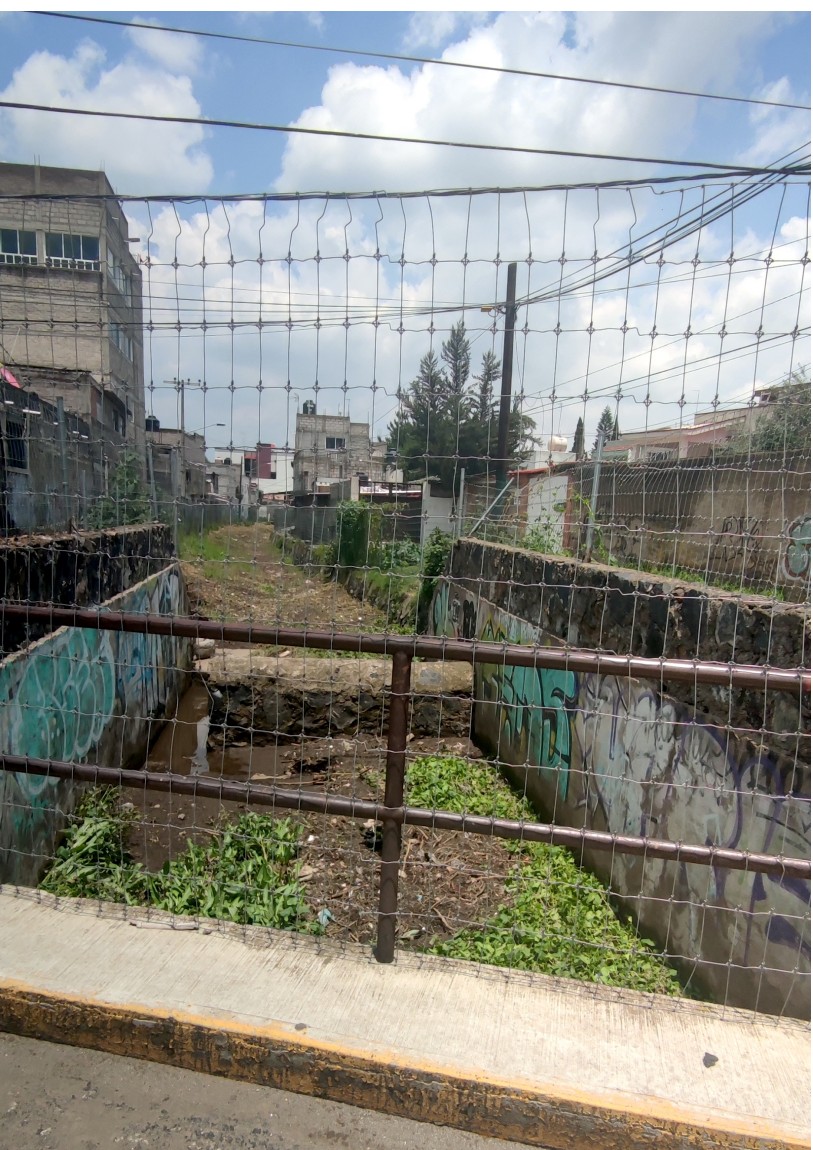

**Figure 13.** Barranca seca in the San Antonio Tecómitl town.

Due to the effects deriving from the overflowing of the Barranca Seca, the population of San Antonio Tecómitl has autonomously built elevations at the entrance to the roads to prevent the entry of water during the rainy season, which runs from July to September (Figure 14).

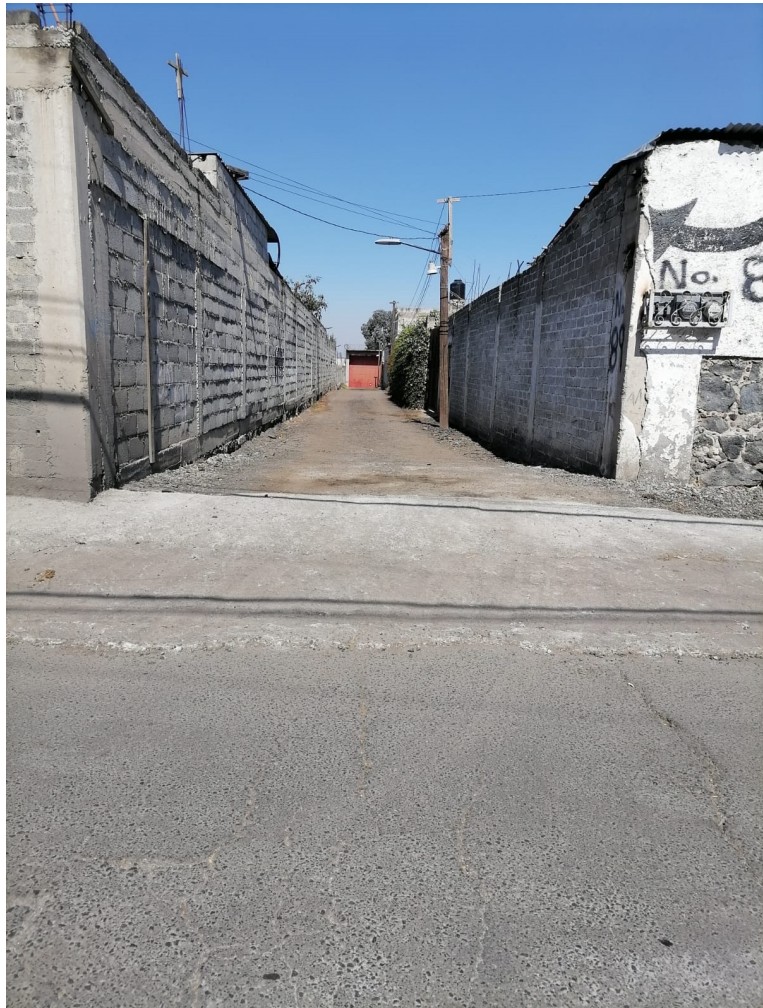

**Figure 14.** Elevation built at the entrance to a road to prevent the influx of water during the rainy season.

The flooding scenario here depicted is commonly found worldwide in plain areas located at the foot of steeper slopes, which have undergone wild recent urbanization superimposed on the natural stream network [55,56]. Under this scenario, urbanized areas act as a barrage for the surface runoff, creating the triggering conditions for flooding. Adopting mitigation strategies able to reduce the flooding risks is not so easy, also due to the intensification of extreme meteoric events caused by climatic changes. The only effective solution is the reinstatement of the pristine stream network, now replaced by artificial drainage channels adopting hydraulic sections designed to cope with the new expected maximum rainfall intensities, generated by climatic changes. From a socioeconomic point of view, this hydraulic-effective solution is not easy to adopt. It is costly and has a negative impact on the population, due to the need to demolish buildings that could impede the correct implementation of the drainage channel network. On the other hand, adopting less impactful solutions, such as taking care to keep existing channels free of vegetation, waste, and debris in order to favor the fast passage of surface runoff or elevating the banks of the channels for coping with high hydrometric levels following intense rainfalls, could result in only a palliative cure not able to solve the problem.

The construction of stormwater detention ponds (SDP) in the upper reaches of the catchment area upstream of urban areas is a well-established technology for reducing the flood risk of urban areas due to flash floods [57–59]. Naturally, the SDP project needs many more in-depth studies regarding the infiltration capacity and hydrological dynamics of the different land uses in the different soils of the study area. Compared to the previous studies

available in this area, a further level of detail is required for the SDP project. Furthermore, it is desirable that a multiplicity of SDPs be created, also distributed on the same flow line, in order to increase the water storage capacity and guarantee higher levels of safety [59]. The water that can be stored could be used for irrigation or, after possible treatment, reintroduced into wells to replenish deep aquifers.

The central portion of the study area is occupied by non-urbanized land (the "undefined" soil as classified by INEGI) mostly in a state of abandonment, affected by bare land or with spontaneous vegetation and, in small part, by nopal crops. The results of the hydraulic simulations in this part of the calculation domain, which take into account the low soil infiltration potential, show that runoff here is already present starting from relatively low return periods, with depths of an order of 40 cm. These levels increase with an increasing return period, exceeding 55 cm. In certain portions of this land, it is possible to note, for the 500-year return period, specific discharge values > 0.60 m$^2$/s (Figure 10), which may have an important erosion effect considering the degree of abandonment of this rural land. The current state of land use in this area also determines the likelihood of flooding in the downstream urban centers, with water depths exceeding 1.50 m.

It is important to consider that, according to information provided by INEGI [60], this part of the study area crosses the region of high recharge potential of the aquifer (Figure 15). However, given the current conditions of land abandonment and the runoff values calculated for the different precipitation scenarios considered it is very probable that water infiltration is prevented or very limited.

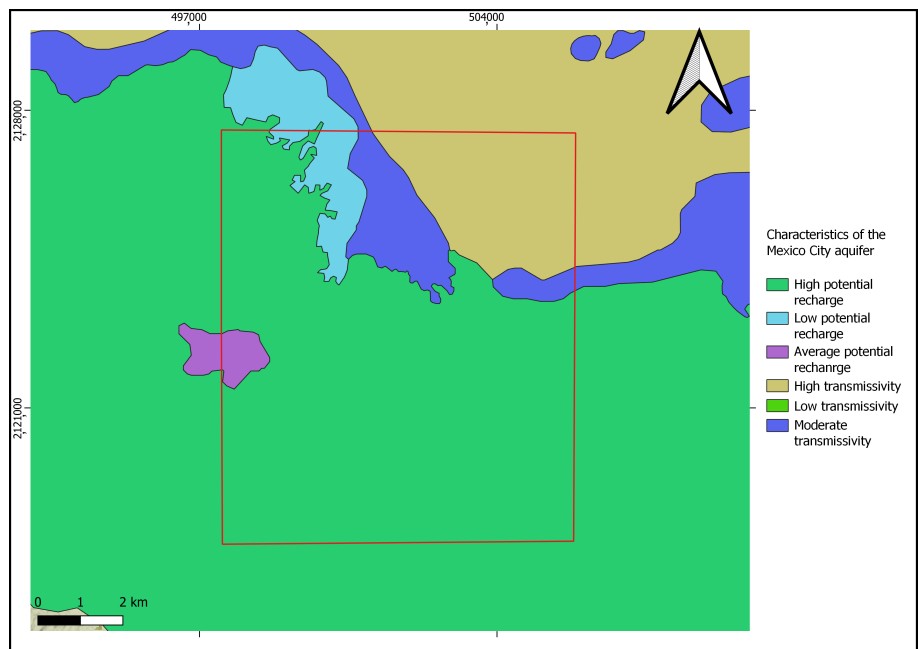

**Figure 15.** Characteristics of the aquifer of the Metropolitan Area of Mexico City. After [60].

To verify the effect that an improvement of land use in this zone can have, an additional simulation was carried out for the 500-year return period by changing the curve number of the "undefined" land from 80 to 69. The result shows that the runoff in this area completely disappears, and the flooding levels in the downstream town are reduced by more than 1 m. This could have a major effect on aquifer recharge if the area was dedicated to crops that favor the deep infiltration of water. This area is already occupied for a small part by nopal cultivations, which are generally characterized by having a superficial and fleshy root system that is distributed horizontally and whose extension depends on the type of soil used [61,62]. Under favorable soil conditions, these roots can penetrate up to approximately 30 cm and may have the ability to cause rapid and significant absorption of rainwater, reducing runoff [63]. Furthermore, the aerial structure of the nopal is also able

to intercept large quantities of rain, which are transferred to the ground as a vapor flow, increasing infiltration and reducing erosion [64]. An intensification of nopal cultivation in this area could therefore lead to a considerable reduction in runoff, as shown by the result of the simulation carried out by lowering the curve number of this type of soil. However, thorough research on the actual ability of nopal to promote deep water infiltration for possible aquifer recharge should be conducted.

A solution for both controlling the runoff excess following intense rainfalls and ensuring an additional recharge of groundwater bodies, aiming to reduce the water table depletion triggered by aquifer overexploitation, could be the creation of temporary water accumulation basins feeding injection wells to deep aquifers. An exhaustive overview of such systems is contained in a report from USGS [65]. The reinjection of the runoff excess could be an effective solution to these problems, but its implementation should be based on very detailed hydrogeologic studies aimed to individuate and solve the numerous related criticalities, in particular: (i) finding free areas to locate water accumulation basins large enough to contain the runoff excess generated by the most intense rainfall; (ii) controlling the quality of the accumulated runoff, in order not to pollute groundwater bodies with contaminants dispersed on the ground and dissolved/suspended by surface waters; and (iii) controlling the suspended solid transport, in order to avoid the fast filling of the accumulation basins and mitigate the risk of occluding the injection wells with sediments. A study by Ghiglieri et al. [15], with the aim of identifying the areas with the greatest intrinsic recharge capacity of the aquifers, could be integrated with studies of the connectivity of the flows that would allow the areas associated with the greatest level of connection with the main drainage lines to be identified [66–68]. These identified areas with greater connectivity could be associated with pond harvesting systems (PHS) [69,70], smaller in size than SDP systems, which, in any case, can make a contribution sensitive to the reduction of local outflow peaks.

## 5. Conclusions

This study has proposed an analysis of the hydraulic structure of an urban area where the expansion of urban settlements, outside of a correct territorial planning, has led to a drastic and negative land use change. The study was applied to an area southeast of Mexico City, classified as conservation land, downstream of a mountain system and rich in dense vegetation. The study area mostly includes the Milpa Alta delegation, part of the Tlahuac delegation and the State of Mexico, and represents one of the areas of Mexico City most affected by flooding during the rainy season, especially in conjunction with particularly intense precipitation events due to extreme weather phenomena. This area also has the particularity of being located in a recharge zone of the aquifer that supplies water to the city and where a high density of water extraction wells is concentrated. The applied methodology, which combines a hydrological approach based on the analysis of available precipitation data and a hydraulic approach that exploits land use data available in the national database, has made it possible to calculate the runoff in terms of water depths and specific discharges on the basis of reconstructed precipitation scenarios corresponding to four return periods.

The results obtained allowed the main hydraulic problems of the urban area to be identified: (i) natural canyons descending from the mountains in Milpa Alta had been transformed into cemented channels with hydraulic sections not adequate to drain high specific discharge rates generated by intense and extraordinary rainfalls, and (ii) the central portion of the study area, occupied by deforested land, in a state of abandonment and affected by waste fires of various kinds, does not promote the infiltration of water and consequently favors runoff, leading to the flooding of inhabited centers in the State of Mexico and in the Tlahuac delegation. This is a preliminary study that aims to raise awareness of the problems related to the formation of flash floods in the region, as well as in all those territories that have characteristics similar to the southeast of Mexico City. Due to its preliminary nature, it contains some shortcomings that can be overcome by making

the following improvements: (i) the land use analysis has to be improved with a precise characterization at a local scale, which can be realized by applying the detailed observation of satellite images; (ii) the hydrological study can be improved taking into account an initial soil moisture condition linked to the historical statistical data of the precipitation; and (iii) the use of a higher resolution digital elevation model in hydraulic simulations would help in the precise localization of the most critical channels and therefore in the more precise calculation of the critical discharge values that cause the overflow and flooding of the residential area.

This study has led to the identification of a fairly common problem in areas characterized by arid or semi-arid climates, prone to flash floods triggered by intense rainfalls, and affected by recent and rapid urbanization processes, not developed following sustainable hydraulic criteria. Actions to be undertaken to reduce runoff are not simple and require a considerable economic inversion. However, this is an extremely environmentally important portion of Mexico City where investing for the improvement of the current situation, in terms of runoff control as well as aquifer recharge, is an action that must be taken into consideration. The inversion, as well as economics, must also be of an educational nature, aimed at providing the right information and the right elements to the population to understand the importance of the area they inhabit and create knowledge regarding the usefulness of possible hydraulic structures, such as rainwater accumulation basins, which are not only useful for reducing the risk of flooding in urban areas, but can also be used to feed injection wells for recharging aquifers. The action of informing the population must also be aimed at creating new habits for the sustainable use of areas that have not yet been urbanized. For example, the results of the simulations in the central portion of the study domain, occupied by bare land and for a small part cultivated with nopal, showed that a change in land use, adopting crops that favor infiltration, would be able to completely eliminate excess runoff and significantly reduce flooding in inhabited areas.

**Author Contributions:** Conceptualization, R.B., L.B. and P.M; methodology, R.B.; software, R.B.; validation, R.B.; formal analysis, R.B., L.B. and P.M.; investigation, R.B., L.B. and P.M.; resources, R.B.; data curation, R.B.; writing—original draft preparation, R.B.; writing—review and editing, R.B., L.B. and P.M.; visualization, R.B.; supervision, R.B.; project administration, R.B. All authors have read and agreed to the published version of the manuscript.

**Funding:** This research received no external funding.

**Institutional Review Board Statement:** Not applicable.

**Informed Consent Statement:** Not applicable.

**Data Availability Statement:** Not applicable.

**Acknowledgments:** The authors would like to thank Karen Montiel and her family for the help offered during the field survey and the experiences recounted concerning the flood events that occurred in their territory.

**Conflicts of Interest:** The authors declare no conflict of interest.

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
