# Peer review of "Analysis of Flow and Land Use on the Hydraulic Structure of Southeast Mexico City: Implications on Flood and Runoff"

_land, doi:10.3390/land12061120_

Round 1
Reviewer 1 Report
Dear Editor.
I have finished my review on the proposed paper “Analysis of flow and land use on the hydraulic structure of southeast Mexico City: implications on flood and runoff”, land-2370428-peer-review-v1.
Summary of the manuscript:
In the proposed paper, the authors’ goal is to simulate the flood discharge over the area of southeast of Mexico City. They used a combination of hydrological analysis and hydrodynamic simulations based on the current land use. They also examined the influence of current land uses and urban sprawl. The results showed that there is an urgent need to adopt actions to reduce flooding and favor infiltration in the area of the city that is also important for the aquifer recharge.
General review:
1. Generally, the manuscript presents a very interesting topic and the specific research seems to include some significant points for the research community of this field.
2. The proposed paper is written with good use of English language. Except some very minor grammatical mistakes and word errors. The authors should check again the paper to correct these minor mistakes.
3. The structure of Introduction is little strange. It seems more as study area description (see below comments).
4. The methodology is generally very interesting, and well explained, so other researchers could easily repeat it. There are some issues that need clarification (see below comments).
5. The results are generally OK, except some points (see below comments).
6. The quality of the work in Discussion I think that is OK.
7. Conclusions are appropriate for this paper.
Additional points for revision:
In my opinion, the proposed paper could be characterized as a good research work, complies with aims of LAND. However, I have few major comments.
INTRODUCTION: The authors begin the paper with the Introduction. In Introduction the reader expects to see the state-of-the-art of the specific issue. Previous studies of how other researchers approached the problem. However, in this Introduction, we read an extended description of the study area. Only between 77-98 lines we can find typical Introduction section. I propose to reorganize this section. Move the description of the study area to “2.1. Description of the study area” and write a normal Introduction section.
Line 137: “….the conurbated area…”. I think “urban area” is better.
Lines 161-166 and table 1: How you calculated the area of influence (Ai) for each station? You did not refer anything about this.
Line 223-225: What about the initial soil moisture conditions? Did you take it into account? Initial soil moisture conditions (or Antecedent Moisture Conditions, AMC) are very crucial for the hydrological modeling.
Lines 241-247: Here, and generally in the text, you refer some extreme flood events of the past. Did you try to calculate the return period of these events and compare your results with these events? With this way you could validate yours results.
Lines 249-254: These lines should be moved to material and methods section.
Lines 259-264 and GENERALLY in the text: I do not understand the measurement unit of specific discharge, m2/s. What you mean saying that the discharge of the water in 0.63 square meters m2 (!) per second? As far as I know the unit of specific discharge is m3/s/Km2 or m3/s/m2. I search for the m2/s, but I found that this unit is used some times for the underground flow. You should change the measurement unit. Also, correct the figures 11 and 12, accordingly. You can not compare your results with other studies when using this unit.
Lines 297-319: You went to some canals in the city. Why you didn’t measure the dimensions of these hydraulic structures, in order to calculate the maximum water capabilities? With this way you could examine if the calculated flood discharges could flow into these structures. Without specific field measurements of these structures you fall into speculations line the lines 316-319.
Lines 324-326: Please, add here the following studies (doi.org/10.3390/hydrology8040170 and doi.org/10.1016/j.apgeog.2018.07.022).
I have commented on the quality of english writing in my review.
Author Response
Dear Reviewer,
We have carefully reviewed the comments and have revised the manuscript accordingly. Our responses are given in a point-by-point manner in the attached file. Changes and correction to the manuscript are highlighted in bold.
Sincerely,
Rosanna Bonasia

Reviewer 2 Report
Review report of the paper "Land 2370428"
I have reviewed the manuscript entitled "Analysis of flow and land use on the hydraulic structure of southeast Mexico City: implications on flood and runoff". I have found the manuscript interesting, addressing an important topic, and using an appropriate methodology, which is well supporting the results. My opinion is quite positive on considering the manuscript for publication in the Journal "Land". However, I believe that there are many major/minor specific comments (listed below) to be addressed first before considering a total acceptance. I do expect the authors to return point by point detailed answers to these questions.
My final decision is “major revision”.
General comments
There is a small problem in your reasoning for land use in this work. You say that all the analyses you do relate to current land use. But, in many places in the text, you speak about changes in land use modes, without you having demonstrated beforehand that there has been a change compared to an earlier date. Nor do you allude to authors having worked in the area who would have made such observations. Consider solving this problem. Either you make classifications of satellite images at several dates to demonstrate that there have been changes in the land use modes, or you call on recent work that would have demonstrated this.
If you insist on talking about land cover change throughout the text as you do, the model must first be made at an earlier date (old period). This would make it possible to make a comparison with the results obtained with the occupation of the current land use mode (what you have in this work). In this way, you will be able to demonstrate that changes in land cover have an impact on the hydraulic structure of the area studied. The proportions of this impact could even be quantified.
What do you mean by green areas? In my opinion, green areas refer to areas with vegetation. Spaces with vegetation always favour infiltration more than those without. When you say that "Runoffs are also intensified by the presence of green areas in a state of abandonment, 10 whose soil does not favour infiltration and promotes flooding of urban areas with water levels higher 11 than 1 m" (lines 10 and 11) ca not understandable. What type of occupation can play such a role? I do not think so. Consider reviewing your reasoning.
Abstract
Finally, some key metrics/values should be provided within the abstract.
Introduction
You choose to begin the presentation of the study area in the introduction (lines 20 to 43). It shouldn't be done in a scientific paper. As a general rule, the presentation of the study area is done exclusively in the first subsection of the "Data and Methods" section. So try to follow the rule. You can briefly mention the study area in the introduction and say what motivates you to study it. But you can't put all these details in the introduction of an article.
The topic on which is sufficiently documented. But you just cite 13 works in your introduction. Just 3 of these works were published after 2015. So try to consolidate your introduction further by including more recent articles (published after 2018) that deal with the subject. Your introduction should have at least 20 citations.
Materials and methods
Your Figure 2 is inappropriate. You call it a "Physiographic map", which does not correspond to anything. In "physiographic", there is relief, soils, geology, etc. Your card does not include these elements. It seems to me that you want to present through this map the hypsometry of the region. We still see the units (mountain, plateau, etc.), but we have no idea of the altitudes. It would be easier to take a digital terrain model, and then make a hypsometric map that will give us better information on the topography of your study area.
In Table 2, is it "soil type" or "land use mode"? I think it's "land use mode"
Why do you call subsection 2.2 "Hydrological study for the definition of precipitation scenarios" when you only deal with precipitation. It seems to me that "Hydrological study" has no place in this title.
The description you make (lines 161 to 165) needs to be reviewed. You speak about "basin" (line 164) when you are not working in a basin.
Is the land cover map produced by INEGI that you use reliable? What year is this card? It is clear that there is a large part of the study area which is classified as "undefined". The problem with this type of information is that it is often designed for very large spaces and it is very imprecise. I sincerely believe that this may influence any interpretations you make from this information. Unless you think that land use does not have a big influence on what you do for work. In such works, it is always preferable to make a supervised classification from satellite images. This allows you to have a map that is closer to reality.
Author Response

(The authors gave the same response as above.)

Reviewer 3 Report
Review of manuscript "Analysis of flow and land use on the hydraulic structure of southeast Mexico City: implications on flood and runoff" (land-2370428)
Dear authors, your research aims to combine the hydrological analysis for the definition of precipitation scenarios, with hydrodynamic simulations based on the current land use in the southeast of Mexico City. It is a well-prepared manuscript and fits the aims and scope of the journal topic. Nevertheless, the authors need to highlight the soundness and novelty of their research as compared with previous research. Therefore, "Major Revision" is necessary to improve this manuscript. Specifically, the reviewer has the following comments and suggestions:
(1) It is unreasonable that the authors directly mentioned the detailed conditions of the study area at the very first parts of the Introduction Section. These parts should be mentioned in the second section, i.e., Materials and Methods. In fact, these parts should be substantially condensed into one or two small paragraphs.
(2) Figure 1. Location map of the study area, delimited by the red line: does it suggest that your study area is exactly the same with the areas of the red rectangle? Why not consider the administrative boundaries?
(3) The Abstract and Introduction Section: overall, these two parts are not strong because the authors did not highlight the necessity and novelty of this study from an international perspective. As a consequence, reviewers cannot figure out why this research must be performed in this context. If this research just presented a case study in a particular region, namely, the southeast of Mexico City, then it lacks enough novelty for publishing in this internationally distinguished journal (Land). I would like to remind that these several flood models were not new in flood risk or runoff assessment.
(4) At the end of the Introduction, the authors have mentioned several studies related to flood risk assessment, but without mentioning the disadvantages of these studies (below).
https://doi.org/10.1016/j.jhydrol.2014.09.054
https://doi.org/10.1007/s10040-016-1427-6
(5) Although this manuscript emphasizes the influence of land use, I am still missing the detailed land use map of this study area.
(6) The Literature Part: in this part, the authors need to look further into the relevant research about future flood risk estimation. In particular, some advanced methods have been largely used in flooding susceptibility assessment (please find below). Nevertheless, these new methods were ignored in this manuscript. A thorough literature review is meant to set the context for your research work and highlight how it contributes to the knowledge in this field and builds on previous relevant research.
https://doi.org/10.1016/j.scs.2022.103812
https://doi.org/10.1080/10106049.2017.1316780
(7) Line 157-162: The precipitation values used to calculate rain intensities were obtained from the hydrometherological stations closest to the study area. But where are these stations, and how many? Please clarify this information.
(8) Line 213-220: The authors should clarify how to determine the curve numbers for those different land use types. Similar questions to the Table 3. Numerical parameters for hydraulic simulations.
(9) Section 2 Materials and Methods: overall, in this part, the authors did not present all the details of the input data, such as the dates in acquiring them, accuracies, temporal and spatial resolutions. The authors should largely improve their statements and expressions.
(10) Figure 12. Specific discharge values calculated for the return period 500 years: please clarify why the specific discharge value results did not match with the land use background.
(11) The authors also need to improve the Conclusion part by clarifying the main shortages of your work.
Moderate editing of English language.
Author Response

(The authors gave the same response as above.)

Round 2
Reviewer 1 Report
Dear authors.
Thank you for the provided responses to my comments. Concerning the Introduction, I believe that the newly form is much better from the the previous. I do not have other comments for the Introduction. In lines 203-204 (revised text) you added the Thiessen method. Please, add here a reference about this method. Please add the following studies (doi.org/10.1002/hyp.13913 and doi.org/10.1016/j.ijdrr.2018.10.015)in lines 381-385 (revised text) to enhance the discussion section. Concerning the responses about the units of specific discharge and the calculation of the discharge capabilities I am fine. This work has good potentials for a new study in the future.
The english language is OK. A last a check for minor word errors.
Author Response
We thank the reviewer for his last suggestions to improve the paper.
We have indicated the bibliographic reference to Thiessen's work of 1911. Regarding the suggested citations, we consider that the first is appropriate to be included in the introduction of our work.
Best regards.
Rosanna Bonasia
Reviewer 2 Report
The authors answered almost all of my concerns. Their article can be published now. In the abstract, I saw that the flow is in m2. I think it should be in m3.
Author Response
We thank the reviewer for the final positive comment on our work.
Regarding the unit of measurement of the flow, we specify the following: In this work the concept of Specific Discharge is used.
The specific discharge describes the variation over time of both the velocity of the flow and its depth.
The equation q = h⋅u, represents the relationship between discharge, flow depth, and velocity in a river or stream. It states that the discharge (q) is equal to the product of the flow depth (h) and the velocity (u), where the units of q are cubic meters per second per unit of area (m^3/s/m). If we assume that the variation in time of both the velocity and flow depth can be roughly estimated as a linear function, then we can represent the graphs of these variables as triangular shapes. This is because a linear function produces a straight line on a graph, and the area under the line represents the total value of the variable over a certain period of time.
Reviewer 3 Report
I appreciate the authors' efforts to improve this manuscript. Now it is acceptable for publication.
Author Response
We thank the reviewer very much for his final comment.